# Long term prophylactic anticoagulation for portal vein thrombosis after splenectomy: A systematic review and meta-analysis

**Zheng Liao, Zixiang Wang, Chenguang Su, Yinxuan Pei, Weiwei Li, Jinlong Liu***

Department of Hepatobiliary Surgery, Affiliated Hospital of Chengde Medical University, Chengde, Hebei, China

* liujl800813@163.com

## Abstract

### Aim

The aim of this study was to evaluate the efficacy and safety of the anticoagulants for the prevention of portal vein system thrombosis (PVST) in patients with cirrhosis after splenectomy and explore the optimal time of anticoagulant administration.

### Methods

A systematic literature search was performed using PubMed, Embase and China Biology Medicine disc (CBM)databases, so as to screen out studies comparing the prognoses between cirrhotic post-splenectomy patients treated with and without anticoagulants. The parameters that were analyzed included the incidence of PVST and postoperative bleeding.

### Results

With a total of 592 subjects, we included 8 studies (6 observational and 2 randomized trials) that fulfilled the inclusion criteria. We found that the incidence of PVST was significantly lower in the anticoagulation group during the first 6 months of anticoagulant administration. And the largest difference in the incidence of PVST between the anticoagulation and control groups was observed at 3 months (odds ratio 0.17(0.11~0.27); P = 0.767; I2 = 0.0%) and 6 months (OR = 0.21(0.11~0.40); P = 0.714; I2 = 0.0%) postoperatively. The incidence of bleeding was not significantly higher in the anticoagulation group (odds ratio 0.71 (0.30~1.71); P = 0.580; I2 = 0.0%).

### Conclusion

Low-molecular weight heparin (LMWH) and warfarin can decrease the incidence of PVST in post-splenectomy cirrhotic patients without an increased risk of bleeding. And the optimal use time of warfarin is 6 months after splenectomy.

**Data Availability Statement:** All relevant data are within the paper.

**Funding:** This work was supported by the Hebei Provincial Key Research Project (21377767D).The

funders had no role in study design, data collection and analysis, decision to publish, or preparation of the manuscript.

**Competing interests:** The authors have declared that no competing interests exist.

# 1. Introduction

Cirrhosis is one of the increasing causes of morbidity and mortality, being the 14th most common cause of death worldwide [1]. Cirrhosis can increase portal vein pressure and lead to hypersplenism, and splenectomy is often used to treat patients with portal hypertension and hypersplenism [2]. Since splenectomy is widely used due to the large number of patients with hepatitis B, the management of post-splenectomy complications is particularly important. And the most serious complications are post-operative infection and portal vein system thrombosis (PVST) [3].

PVST refers to the blood clots that formed in the portal vein, splenic vein, superior mesenteric vein or intrahepatic portal vein branches [4]. There are some studies have shown that the incidence of PVST in post-splenectomy patients ranges from 5% [5] to 52% [6]. The clinical manifestations of PVST include fever, abdominal pain, nausea and vomiting, which may lead to intestinal infarction in severe cases [7]. There is still controversy as to whether anticoagulants should be applied early after splenectomy to prevent the formation of PVST. Some argued that patients with cirrhosis had an increased risk of bleeding due to coagulation disorders [3, 8] and should not be given prophylactic anticoagulation. But some studies [9, 10] showed that both pro- and anticoagulation elements were reduced in cirrhotic patients, which maintain the complex coagulation homeostasis. Thus, anticoagulants can't increase the risk of bleeding.

Therefore, the aim of this meta-analysis was to verify the safety and efficacy of the anticoagulants for the prevention of PVST in patients with cirrhosis after splenectomy. And then we further explored the optimal time of anticoagulants administration to prevent the occurrence of PVST after splenectomy.

# 2. Methods

## 2.1 Literature search

The systematic literature search was conducted by two investigators (Zheng Liao and Zixiang Wang) independently through PubMed, Embase and China Biology Medicine disc (CBM) databases, so as to screen out comparative studies evaluating the efficacy of anticoagulation (with or without anticoagulants) in cirrhotic patients undergoing splenectomy. The search was restricted to comparative studies (prospective or retrospective) written in English or Chinese and published before 31 July 2022. This meta-analysis was performed according to the Preferred Reporting Items for Systematic Reviews and Meta-Analyses statement [11].

The search was based on the following search formula: ((("Splenectomy"[Mesh]) OR (splenectomy)) AND ((("Thrombosis"[Mesh]) OR (thrombosis)) OR (thrombus))) AND (("Anticoagulants"[Mesh]) OR (anticoagulant)). The reference list of screened articles was also checked for potential hits.

## 2.2 Study selection

The eligibility of each study was assessed independently by two investigators. All titles and abstracts had been screened to determine the relevance of each study to this article. Studies meeting the following criteria were included in the meta-analysis: 1. randomized or comparative studies evaluated the effects of anticoagulation (with or without anticoagulants) in preventing PVST in cirrhotic patients undergoing splenectomy, regardless of the numbers of patients in each arm; 2. the dose and timing of post-splenectomy anticoagulation treatment were clearly shown; 3. the information of the incidence of PVST was available.; 4. randomized

controlled trials (RCTs) with an evaluation of four or more 'low risk' and retrospective non-randomized trials (RTs) with a cumulative quality of literature score ≥7.

Studies were excluded if: 1. without a control group; 2. incomplete raw data for the purpose of this research; 3. only animals or cells included in the research; 4. reviews, study protocols, comments or case reports; 5. studies unrelated to the prevention of PVST after splenectomy.

If the two investigators disagreed with the inclusion or exclusion of an article, a meeting was held to determine its eligibility.

### 2.3 Data extraction

The two investigators examined the full text, tables and figures of relevant literature to extract data from articles that have been included in this study. The extracted data included: 1. name of the first author, year of publication, operation method, state of postoperative bleeding and detection medium; 2. demographics (sample size in each group and aetiology); 3. use of anticoagulants in each study 4. primary outcome: post-operative incidence of PVST and secondary outcome: the incidence of post-operative bleeding. The controversies were resolved via consensus between the two authors if they disagreed with the data.

### 2.4 Quality assessment

The methodological quality of RCTs was assessed using the Cochrane Handbook for Systematic Reviews of Interventions [12]. Risk of bias for each eligible RCTs was determined by seven items: 1. random sequence generation; 2. allocation concealment; 3. blinding of participants and personnel; 4. blinding of outcome assessment; 5. incomplete outcome data; 6. selective reporting; 7. other bias. The evaluation results for each item include: 'low risk', 'unclear risk' and 'high risk'. An article was considered high quality if it had three or more 'low risk' and low quality if it has less than three 'low risk'. The methodological quality of retrospective non-RTs was assessed using the modified Newcastle–Ottawa scale [13]. It contains three parts: 1. patient selection (0–4 points); 2. Comparability (0–2 points); 3. outcome (0–3 points). Differences in evaluation regarding the bias of studies were resolved through discussion and consensus.

### 2.5 Statistical analysis

The meta-analysis was performed using stata12.0.The incidences of postoperative PVST and postoperative bleeding were treated as dichotomous data and pooled odds ratios (OR) with 95% confidence interval (CI) were used to perform analysis. The OR is used to reflect differences in exposure between experiments and controls, thus establishing a link between disease and exposure factors. $P<0.05$ was considered to indicate a statistically significant difference between the two groups. In terms of the heterogeneity test, when the statistics $P > 0.05$, $I^2 < 50\%$, it can be assumed that there is no significant statistical difference in the included data, so the fixed effect model is adopted. When the statistics $P \leq 0.05$ and $I^2 \geq 50\%$, indicating that there are significant statistical differences among the included data. The random effect model is adopted. A funnel plot and Egger's test were designed to establish the existence of publication bias [14]. The trim-and-fill method was performed to further assess the potential publication bias.

## 3. Results

### 3.1 Literature search results

According to the preliminary search, the search strategy identified 854 articles, in which 247 articles from the PubMed, 328 from the Embase, 278 from the CBM, and 1 from other sources.

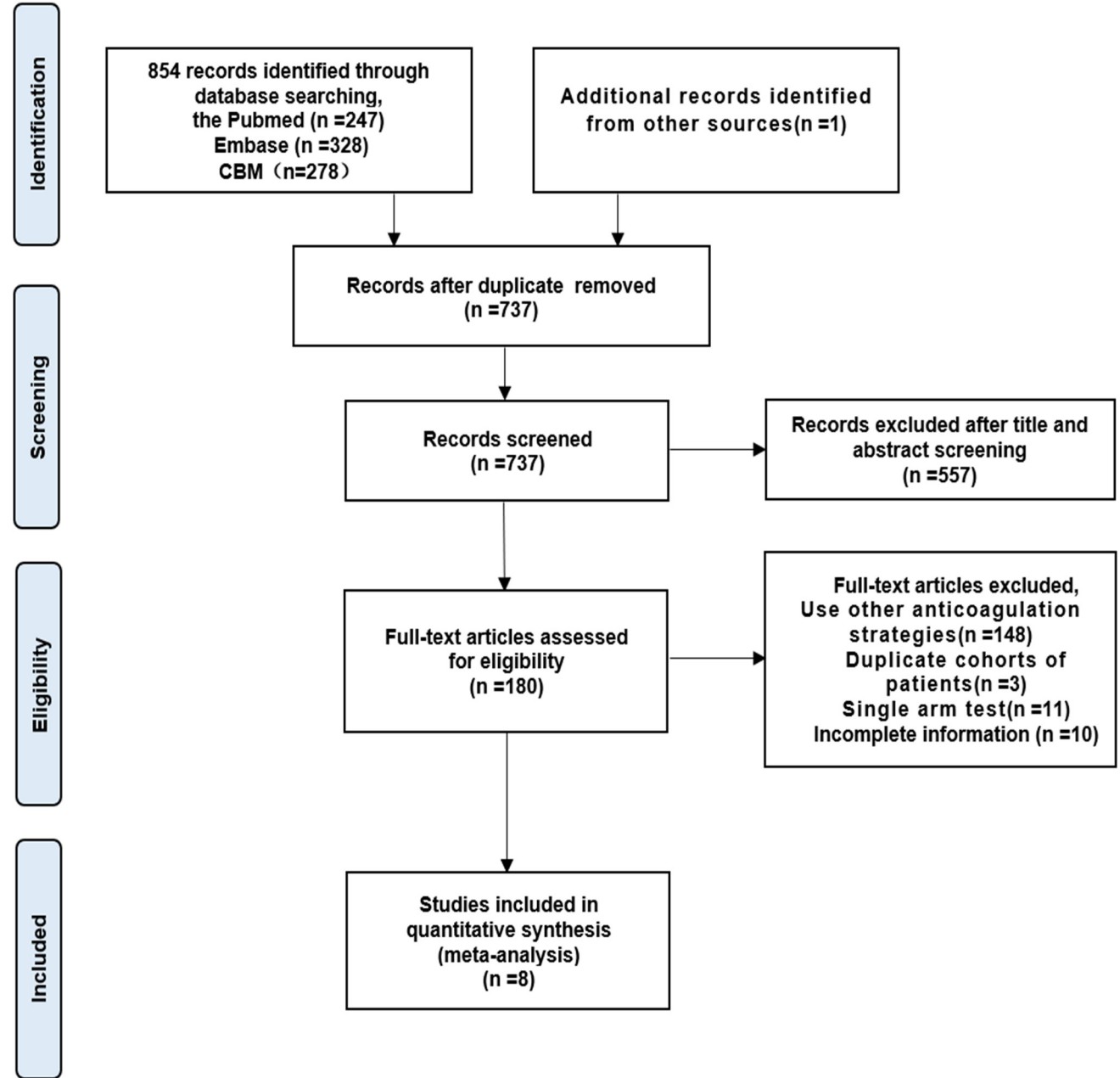

**Fig 1. Flow chart of the study selection procedure.**

Among these, 117 were excluded due to duplication and 557 were excluded after reading the title and abstract, leaving 265 available for further full-text review. After full-text review of the remaining 180 studies, 172 were excluded for reasons shown in Fig 1. Finally, 8 eligible studies [15–22] were finally included in this meta-analysis.

## 3.2 Characteristics of included studies

Two of the eight included studies [15–22] were RCTs, one of which was triple-blinded [17] and the other was single-blinded [16]. The remaining 6 studies [15, 18–22] were retrospective

**Table 1. Characteristics of the included studies.**

| Author, Year | Country | Surgery method | Anticoagulation regimen | | Subjects | | Bleeding(n) Experiment: Control | Etiology | NOS |
|---|---|---|---|---|---|---|---|---|---|
| | | | Experiment | Control | Experiment | Control | | | |
| Li [16] 2022 | China | Laparoscopic splenectomy pericardial disconnection | LMWH,4250U/day POD 1 to 5 warfarin (2.5mg/day) POD 3 to POM 6 | aspirin (100mg/day) on the POD3 for 6 months | 68 | 63 | 1:0 | - | 7 |
| Zhen [15] 2022 | China | Splenectomy azygoportal disconnection | LMWH,5000U/day POD 1 to 7 warfarin (1.5~3mg/day) POD 7 to POM 6 | No prophylactic therapy | 56 | 50 | 2:5 | "HBV; HCV; Alcohol Autoimmunity | 7 |
| Bai [17] 2019 | China | Laparoscopic splenectomy azygoportal disconnection | LMWH,4100U/day POD 3 to 8 warfarin (2.5mg/day) POD 3 to POM 12 | aspirin (100mg/day) on the POD3 for 1 year | 39 | 39 | 4:4 | HBV; HCV; Alcohol Schistosomiasis Autoimmunity Idiopathic cirrhosis | 7 |
| Tang [18] 2018 | China | Laparoscopic splenectomy | LMWH,3000U/day POD 1 to 7 warfarin (3.75mg/day) POD 3 to POM3 | aspirin (100mg/day) on the POD3 for 3 months | 33 | 21 | - | HBV; HCV | 7 |
| Huang [19] 2017 | China | Laparoscopic splenectomy azygoportal disconnection | LMWH,4100U/day POD 3 to 8 warfarin (2.5mg/day) POD3 to POM 12 | aspirin (100mg/day) on the POD3 for 1 year | 31 | 31 | | - | 7 |
| Jiang [20] 2016 | China | Laparoscopic splenectomy azygoportal disconnection | LMWH,4100U/day POD 3 to 8 warfarin (2.5mg/day) POD 3 to POM 3 | aspirin (100mg/day) on the POD3 for 3 months | 34 | 39 | 0:2 | HBV; HCV Schistosomiasis; Alcohol Autoimmunity Idiopathic cirrhosis. | 8 |
| Wu [21] 2015 | China | Splenectomy + pericardial devascularization | LMWH,6000U/day POD 1 to 5 warfarin POD 6 to POM 1 | No prophylactic therapy | 23 | 19 | 1:0 | HBV; HCV | 8 |
| Zou [22] 2013 | China | Splenectomy | LMWH,3000U/day POD 1 to 7 warfarin (3.75mg/day) POD 3 to POM 3 | aspirin (100mg/day) from the POD3 for 3 months | 36 | 20 | - | HBV; HCV | 7 |

"-": the data were not available, POD: postoperative day, POM: postoperative month, HBV: hepatitis B virus-induced cirrhosis, HCV: hepatitis C virus-induced cirrhosis

non-RTs. These 8 studies [15–22] contained 592 patients, 320 of whom were treated with Low-molecular weight heparin (LMWH) and warfarin (experimental group) and the remaining 272 patients were not treated with anticoagulants (control group). The years of publication spanned from 2013 to 2022. The characteristics of each study are displayed in Table 1. All of the patients included in the eight studies [15–22] were Chinese, and they were diagnosed with liver cirrhosis with or without portal hypertension and hypersplenism. The aetiologies of cirrhosis included hepatitis B virus (HBV), hepatitis C virus (HCV), alcohol, autoimmunity, schistosomiasis and Idiopathic cirrhosis.

The results of the quality assessment are as follows: two RCTS obtained more than four 'low risk', six retrospective non-RTs received a score of more than 7.

## 3.3 The incidence of PVST

Patients received continuous subcutaneous injections of LMWH for 5 [16, 17, 19–21] or 7 [15, 18, 22] days and warfarin for 1 [21] to 12 [17, 19] months in the experimental group. And the control group received enteric-coated aspirin tablets for 3 [18, 20, 22] to 12 [17, 19] months or no prophylactic therapy [15, 21] (Detailed information is shown in Table 1).

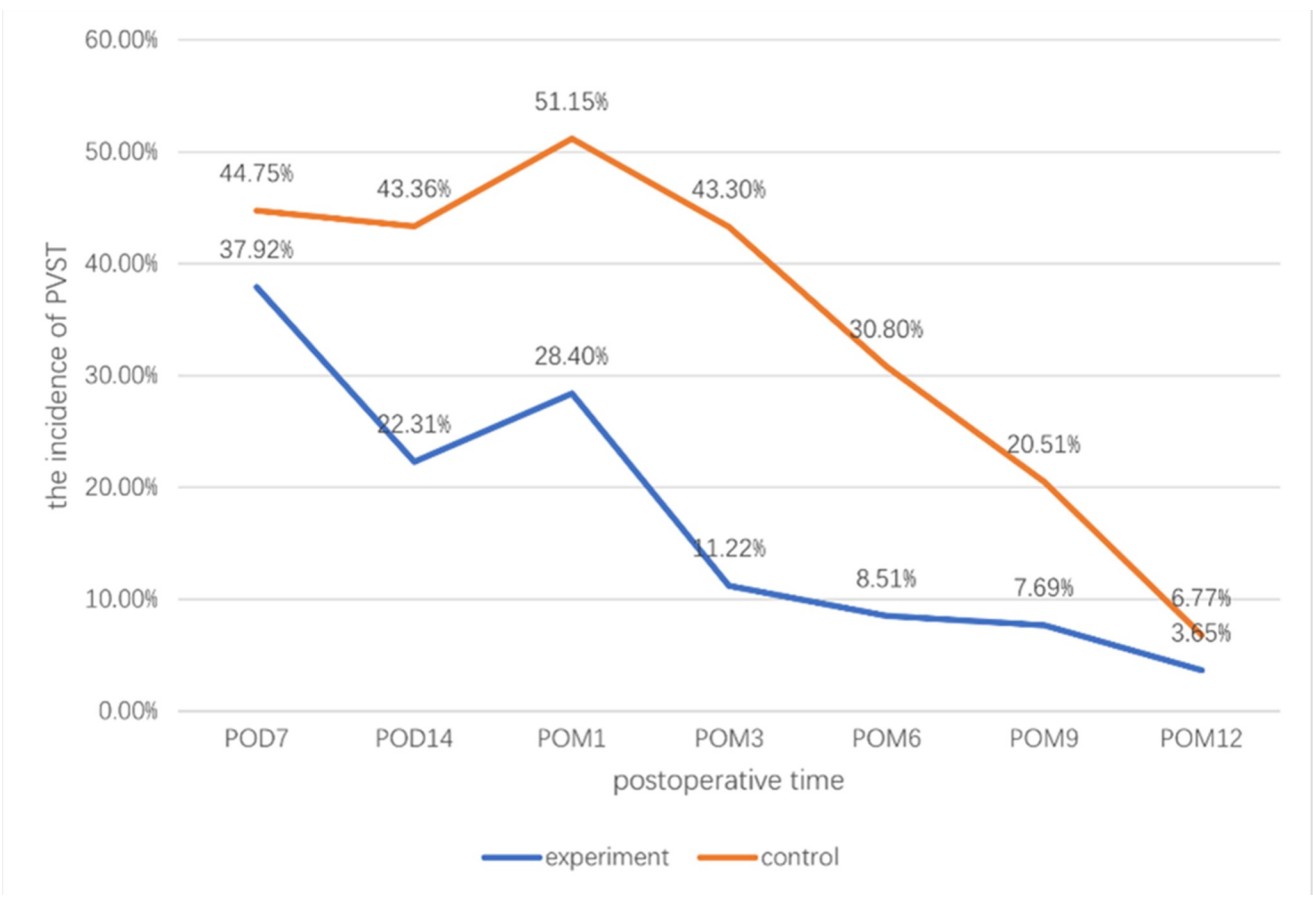

**Fig 2. The line graph of the incidence of PVST.**

In order to investigate the effect of LMWH and warfarin in preventing PVST formation after splenectomy in patients with cirrhosis, we grouped the patients according to their anticoagulation time after splenectomy so as to analyse the incidence of PVST at different times with the use of anticoagulants. Ultrasound and angiography were used in patients at POD 7, 14 and POM 1, 3, 6, 9 and 12 to detect thrombosis in the main trunk or branches of the portal venous system.

We plotted a line graph (Fig 2) based on the incidence of PVST at different times after splenectomy and a subgroup analysis (Fig 3) was performed according to the anticoagulation time. The incidence of PVST decreased first and then increased in the early stage ($\leq$1 month). The incidence of PVST at POD 7 was 37.92% and 44.75% in the experimental and control groups, respectively. There was no statistical difference between the two groups (OR = 0.73 (0.47~1.14); P = 0.799; $I^2$ = 0.0%). The incidence of PVST decreased significantly in the experimental group at POD 14 after splenectomy; The incidence of PVST was 22.31% in the experimental group and 43.36% in the control group (OR = 0.38(0.22~0.67); P = 0.205; $I^2$ = 37.9%). We found an increase in the incidence of PVST in both groups at 1 month after splenectomy, and the rise in the control group was greater. The incidence of thrombosis in the experimental group was 28.40% and was significantly lower than that in the control group (51.15%) (OR = 0.37(0.25~0.55); P = 0.752; $I^2$ = 0.0%). We speculated that this may be due to the high incidence period of thrombosis from POD 14 to POM 1 after splenectomy.

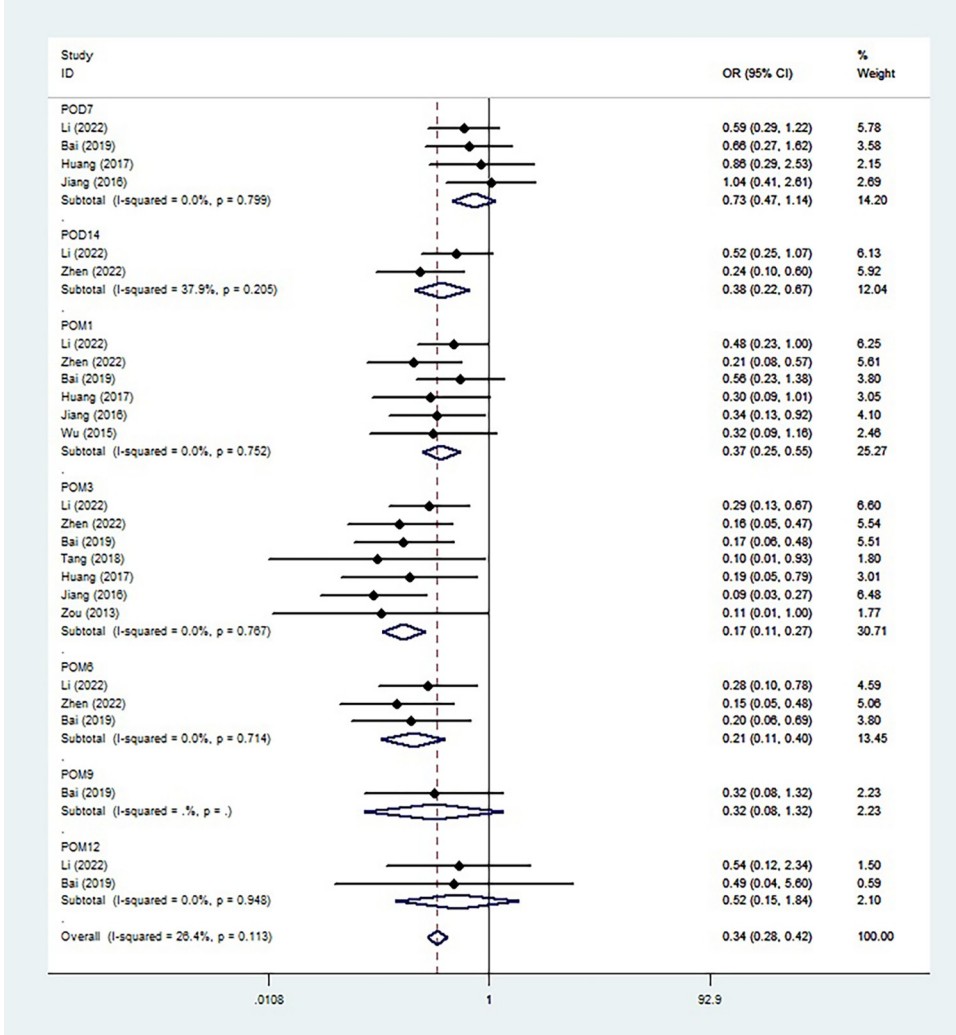

**Fig 3. Meta-analysis of the probability of portal vein system thrombosis following anticoagulants administration after splenectomy.**

The incidence of PVST was 11.22% and 43.30% in the experimental and control groups at 3 months postoperatively (OR = 0.17(0.11~0.27); P = 0.767; $I^2$ = 0.0%). The difference between the two groups at this time was the most significant of all time points, and the incidence of PVST showed a gradual decrease in both groups from 3 months after splenectomy. The incidence of PVST was 8.51% and 30.80% in experimental and control groups at 6 months postoperatively (OR = 0.21(0.11~0.40); P = 0.714; $I^2$ = 0.0%), the experimental group was significantly lower than the control group. The incidence was 7.69% and 20.51% in the experiment and the control groups at 9 months postoperatively (OR = 0.32(0.08~1.32)), and 3.65% and 6.77% at 12 months postoperatively (OR = 0.52(0.15~1.84); P = 0.948; $I^2$ = 0.0%). At the two time points, the incidence of thrombosis was not significantly higher in the experimental group.

### 3.4 Incidence of postoperative bleeding

For the safety of anticoagulants, postoperative bleeding was assessed in the five studies [15–17, 20, 21]. There were 8 and 11 patients occurred postoperative bleeding in the experimental and

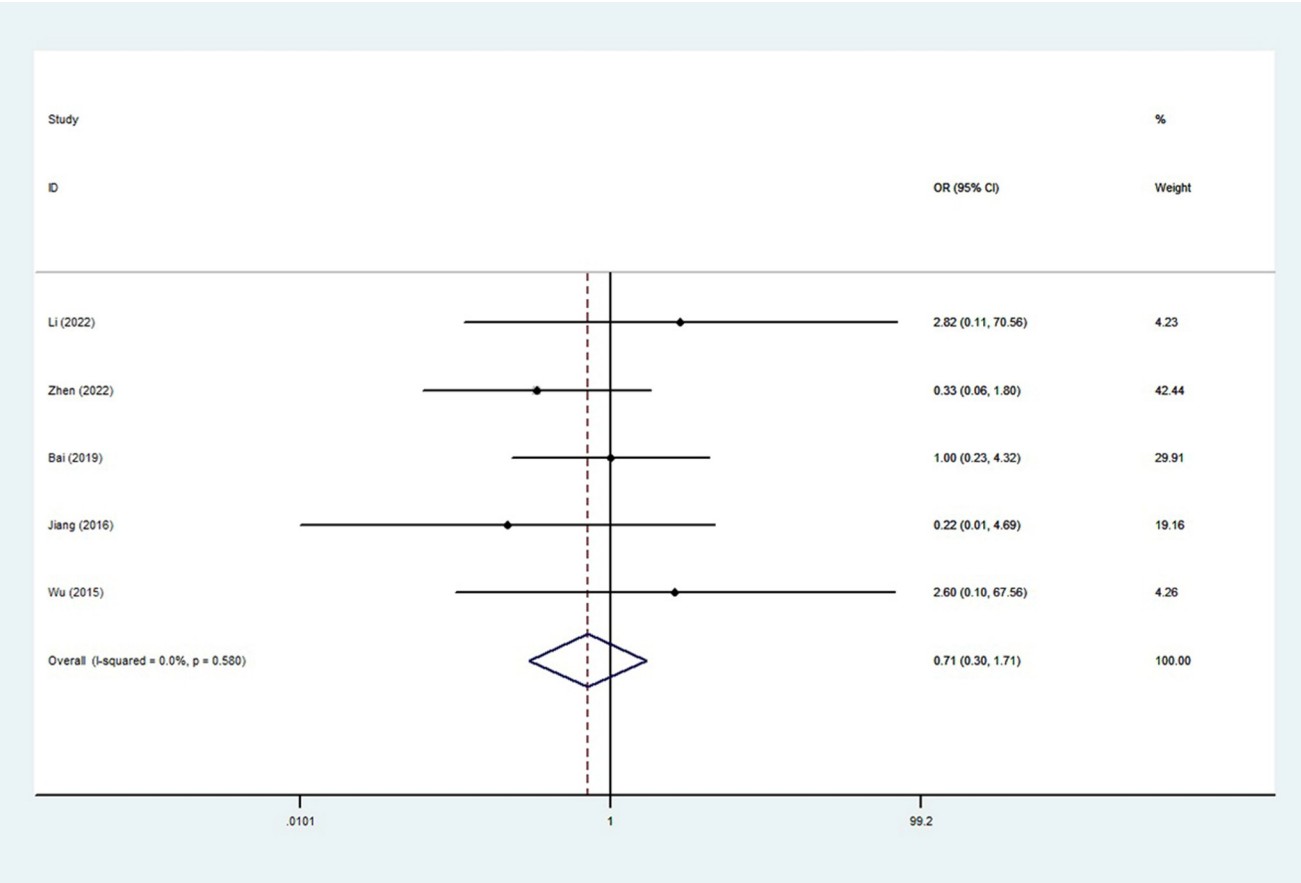

**Fig 4. Meta-analysis of bleeding after splenectomy with anticoagulants.**

control groups, and the incidence was 2.89% and 3.73%. The most common bleeding manifestation in these patients was gastrointestinal bleeding. The incidence of bleeding was not significantly higher in the experimental group(OR = 0.71 (0.30~1.71); P = 0.580; $I^2$ = 0.0%) (Fig 4).

## 3.5 Publication bias

The publication bias was evaluated by funnel plot (Fig 5) and Egger's test. The Egger's test gave a P-value of 0.039($<$0.05) for the incidence of PVST, indicating the potential presence of publication bias. According to the Trim-and-Fill method (Fig 6), the corrected random-effect ORs was- 1.045(95%CI:- 1.252,- 0.838). P-values before and after filling were $>$0.05, there was no substantial difference between the two results. So, it can be assumed that the original results are plausible.

The Egger's test gave a P-value of 0.658 for the incidence of postoperative bleeding (Fig 7), indicating no evidence of publication bias.

## 4. Discussion

The formation of PVST after splenectomy is a complication that should be closely monitored, it can affect the prognosis and even be life-threating. The detailed mechanisms of its formation remain unclear, but it is generally believed to be related to the local hypercoagulable state of the postoperative portal venous system. It may be attributed to the soaring count and

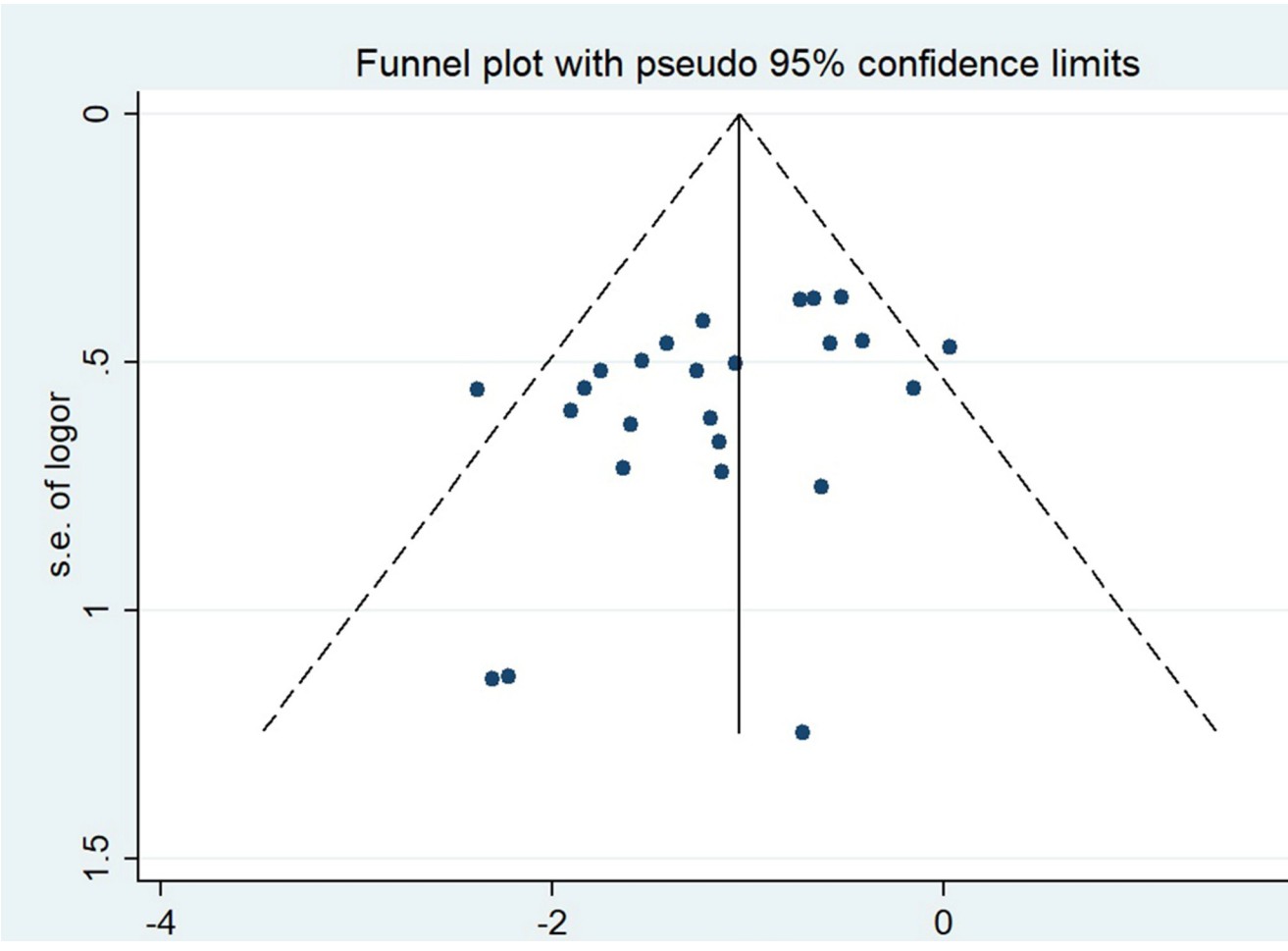

**Fig 5. Funnel plot for the incidence of portal vein system thrombosis.**

augmented aggregation competence of platelets post-surgery. Hemodynamic changes of the portal venous system and the damage of splenic vein may be another important reason for the formation of PVST [23–25]. In addition, the diameter of the splenic and portal veins is thought to be associated with the formation of PVST after splenectomy [26]. Although the prophylaxis of pulmonary embolism and deep vein thrombosis has been relatively well established, but the prophylaxis of PVST after splenectomy remains controversial. Some studies have discussed the risk factors of PVST in patients after splenectomy [23, 24], and indicated anticoagulants could safely and effectively prevent PVST [27–30]. However, these studies didn't describe how the anticoagulants were used. Also, no further studies have been conducted on the optimal timing of anticoagulants use. Therefore, specific anticoagulants and the duration of drug administration were further analysed in our study.

According to the analysis of the eight included articles [15–22], anticoagulants after splenectomy in patients with cirrhosis can effectively prevent PVST. Warfarin works by inhibiting the hepatic synthesis of coagulation factors II, VII, IX and X, which need to wait for the relative depletion of these factors in the body before the anticoagulant effect can be exerted. Therefore, warfarin has a slow onset of action. It should be used in the early days after splenectomy in combination with LMWH to prevent the formation of PVST. Our results showed that the incidence of thrombosis in the experimental group was lower than that in the control group. The

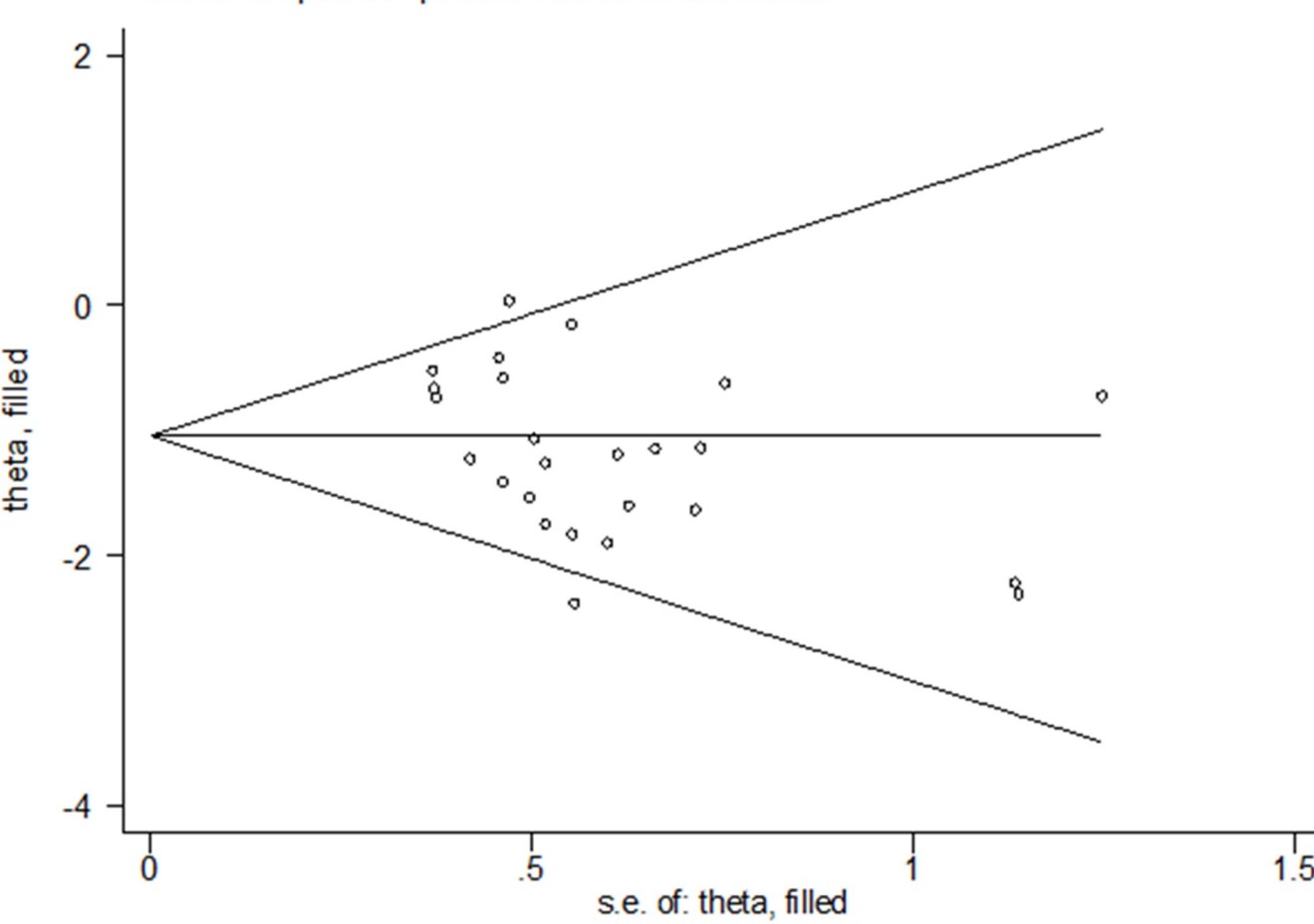

**Fig 6. Filled funnel plot of OR for the incidence of portal vein system thrombosis.**

incidence in experimental and control groups showed a trend of decreasing, then increasing, and finally continuing to decrease. We speculate that this may be due to the high incidence period of PVST from POD 14 to POM 1. The largest difference in the incidence of PVST between the experimental and control groups was observed at POM 3 and POM 6. When the postoperative time was 9 months or 12 months, there was no statistical difference between the experimental group and the control group. This may be due to anticoagulants are not effective for thrombosis older than 6 months. Moreover, the thrombosis of most cases could occur recanalization within the first 6 months of treatment [31]. So, we concluded that the optimal time for warfarin after splenectomy was 6 months.

PVST, a common life-threatening complication after splenectomy in patients with cirrhosis, can lead to a worsening clinical course [32]. The clinical manifestations of PVST include fever, abdominal pain and nausea. It can even cause liver function damage, increase portal hypertension leading to variceal bleeding and increase the risk of ischemic bowel necrosis [33–36]. Furthermore, PVST also may decrease the possibility of liver transplantation in the future [37, 38] and increase the mortality rates after transplantation [39, 40]. From this point of view, the significance of long-term anticoagulants after splenectomy is particularly important. Continuous application of anticoagulants after splenectomy can effectively reduce the occurrence

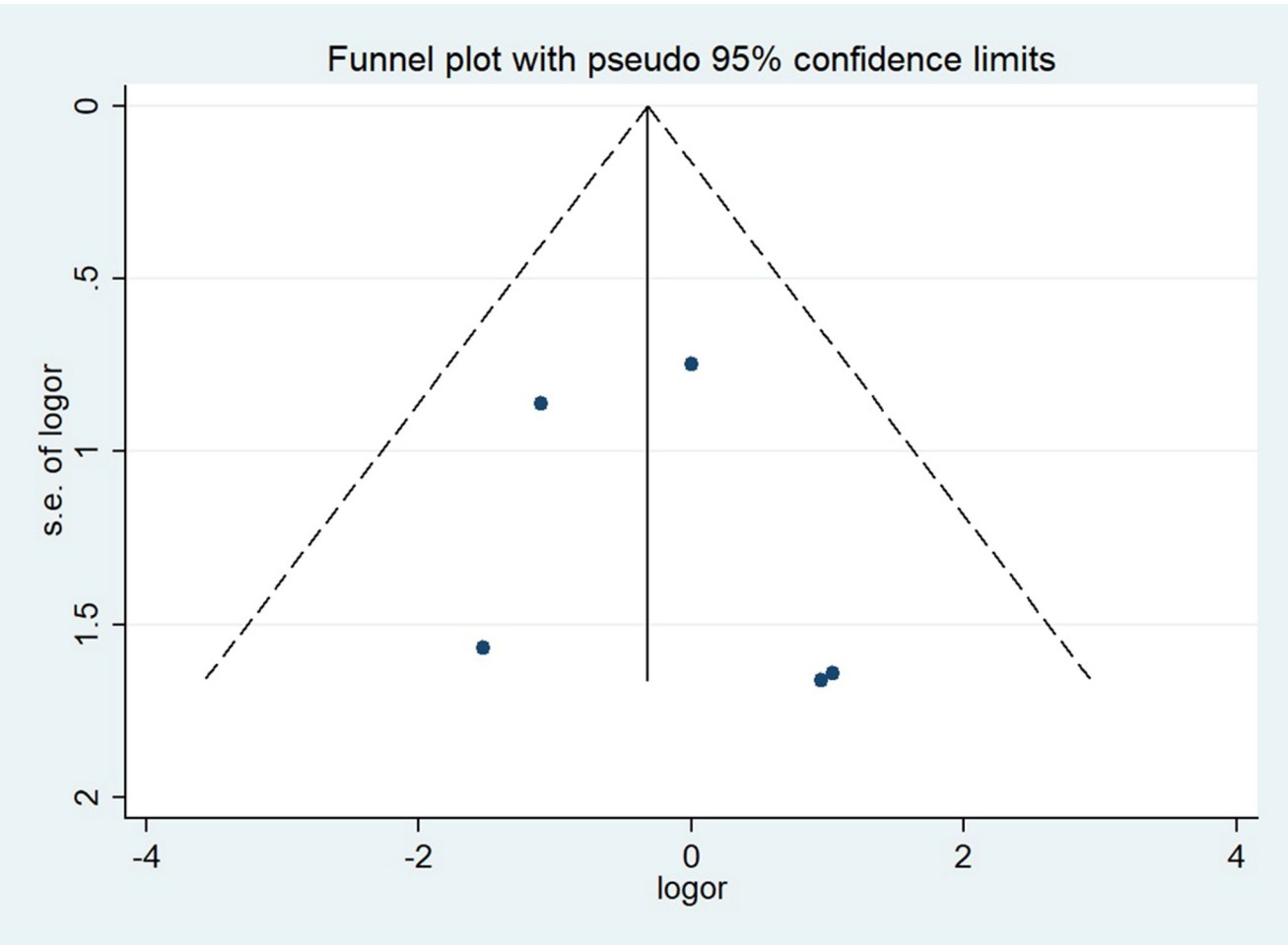

**Fig 7. Funnel plot for incidence of post-operative bleeding.**

of PVST from 1 to 6 months after surgery, prevent related complications, and benefit the prognosis of patients.

A previous meta-analysis [29] suggested LMWH could effectively decrease the incidence of PVST in post-splenectomy patients without an increased risk of bleeding. But it didn't describe how the anticoagulants were used. And there is another RCT [41] reported LMWH was safe and effective for preventing the formation of PVST in patients with cirrhosis. Patients in this study required subcutaneous LMWH at a prophylactic dose (4000 U/day) for 48 weeks. This approach is not only expensive but also requires a high level of patient compliance. Compared to LMWH, warfarin is cheaper and requires only lower compliance for oral administration. In summary, patients with cirrhosis after splenectomy can be treated with warfarin maintenance therapy for 6 months after early postoperative application of LMWH.

Our study had several limitations. First, due to the number of included articles, there is fewer data on long-term warfarin applications in our article, so it is not possible to make a more detailed classification of the duration of warfarin use. Second, the doses of LMWH and warfarin used by patients in the included studies were not identical. Finally, the studies we included were all from China, which may result in a regional bias. Therefore, further studies are needed to overcome these limitations and confirm our findings.

## 5. Conclusions

Our meta-analysis suggested that combined anticoagulation with LMWH and warfarin can decrease the incidence of PVST in post-splenectomy cirrhotic patients without an increased risk of bleeding. Patients after splenectomy can have short term injections of LMWH followed by long term use of warfarin. And patients were best served with warfarin for 6 months after splenectomy to prevent portal vein thrombosis.

## Supporting information

**S1 File.**
(DOCX)

## Author Contributions

**Funding acquisition:** Jinlong Liu.

**Investigation:** Zheng Liao, Zixiang Wang.

**Methodology:** Yinxuan Pei.

**Resources:** Zheng Liao.

**Software:** Zheng Liao, Chenguang Su, Weiwei Li.

**Supervision:** Jinlong Liu.

**Writing – original draft:** Zheng Liao.

**Writing – review & editing:** Jinlong Liu.

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
