## [Decision Letter · Decision Letter 0]

15 Jun 2023

PONE-D-23-13885

The role of anticoagulation in the prevention of portal vein thrombosis after splenectomy in patients with cirrhosis: A systematic review and meta-analysis

PLOS ONE

Dear Dr. Liu,

Thank you for submitting your manuscript to PLOS ONE. After careful consideration, we feel that it has merit but does not fully meet PLOS ONE’s publication criteria as it currently stands. Therefore, we invite you to submit a revised version of the manuscript that addresses the points raised during the review process.

We look forward to receiving your revised manuscript.

Kind regards,

Redoy Ranjan, MBBS, MRCSEd, Ch.M., MS (CV&TS), FACS

Academic Editor

PLOS ONE

Journal Requirements:

   "This work was supported by the Hebei Provincial Key Research Project (21377767D)."

4. Please remove your figures from within your manuscript file, leaving only the individual TIFF/EPS image files, uploaded separately. These will be automatically included in the reviewers’ PDF.

Reviewers' comments:

Reviewer's Responses to Questions

**Comments to the Author**

1. Is the manuscript technically sound, and do the data support the conclusions?

Reviewer #1: Yes

Reviewer #2: Yes

2. Has the statistical analysis been performed appropriately and rigorously? 

Reviewer #1: No

Reviewer #2: Yes

3. Have the authors made all data underlying the findings in their manuscript fully available?

Reviewer #1: Yes

Reviewer #2: Yes

4. Is the manuscript presented in an intelligible fashion and written in standard English?

Reviewer #1: Yes

Reviewer #2: Yes

5. Review Comments to the Author

Reviewer #1: In this study, the authors aimed to assess the effectiveness and safety of anticoagulants in preventing portal vein system thrombosis (PVST) in patients with cirrhosis following splenectomy. Additionally, the study aimed to determine the optimal timing for administering anticoagulants. A comprehensive search of literature databases including PubMed, Embase, and China Biology Medicine disc (CBM) was conducted to achieve this. The search aimed to identify relevant studies that compared the outcomes of cirrhotic post-splenectomy patients who received anticoagulants to those who did not. The parameters analyzed encompassed the occurrence of PVST and postoperative bleeding.

The topic is interesting and falls under the scope of the journal. The article is well organized and well structured. However, the authors need to address the following comments.

Major Comments:

1. Provide the syntax codes of STATA 12.0 in the appendix.

2. Revise section 2.5 and interpret different statistical tools used in this study for the convenience of the readers such as confidence interval, Chi-square test, Egger’s test, etc.

Minor comments:

1. The title is too long for the reader; it should be precise.

2. Table 1 font is too small; readjust to make it more visible.

3. Page 2, line 15, write as “explore the optimal time of anticoagulant administration”.

4. Page 10, line 40, write, “Cirrhosis can increase portal vein pressure and lead to hypersplenism”.

5. The two investigators examined the full text, tables, and figures of relevant literature to extract data from articles that had been included in this study.

6. There are some grammatical mistakes in the article please revise the article thoroughly and remove all the errors.

7. Conclusions are not clearly specified.

8. Some of the references need revision such as reference 7 and 19.

Reviewer #2: The objectives of the study are fully met on behalf of the test results/estimates supported by relevant dataset and statistical analysis. In the light of these findings, I recommend accepting and publishing the article in PLOS ONE.

6. PLOS authors have the option to publish the peer review history of their article (what does this mean?). If published, this will include your full peer review and any attached files.

Reviewer #1: No

Reviewer #2: No

---

## [Author Response · Author response to Decision Letter 0]

22 Jul 2023

Dear reviewers,

Thank you very much for your comments and professional advice. Based on your suggestion and request we have made corrected modifications on the revised manuscript. We hope that our work can be improved again. The details as follows:

Reviewer #1

Major Comments:

1. Provide the syntax codes of STATA 12.0 in the appendix.

The author’s answer: We uploaded my syntax code of Stata 12.0 as supporting information.

2. Revise section 2.5 and interpret different statistical tools used in this study for the convenience of the readers such as confidence interval, Chi-square test, Egger’s test, etc.

The author’s answer: We have revised the section in the article. The details as follows: “The meta‑analysis was performed using stata12.0 .The incidences of postoperative PVST and postoperative bleeding were treated as dichotomous data and pooled odds ratios (OR) with 95% confidence interval (CI) were used to perform analysis. The OR is used to reflect differences in exposure between experiments and controls, thus establishing a link between disease and exposure factors. P<0.05 was considered to indicate a statistically significant difference between the two groups. In terms of the heterogeneity test, when the statistics P > 0.05, I2 < 50%, it can be assumed that there is no significant statistical difference in the included data, so the fixed effect model is adopted. When the statistics P ≤ 0.05 and I2 ≥ 50%, indicating that there are significant statistical differences among the included data. The random effect model is adopted. A funnel plot and Egger’s test were designed to establish the existence of publication bias. The trim-and-fill method was performed to further assess the potential publication bias. ”

Minor comments:

1. The title is too long for the reader; it should be precise.

The author’s answer: We have streamlined the title. The details as follows: “Long term prophylactic anticoagulation for portal vein thrombosis after splenectomy: A systematic review and meta-analysis”.

2. Table 1 font is too small; readjust to make it more visible.

The author’s answer: We have enlarged the font in the table.

3. Page 2, line 15, write as “explore the optimal time of anticoagulant administration”.

The author’s answer: We have revised the sentence. The details as follows: “The aim of this study was to evaluate the efficacy and safety of the anticoagulants for the prevention of portal vein system thrombosis (PVST) in patients with cirrhosis after splenectomy and explore the optimal time of anticoagulant administration.”

4. Page 10, line 40, write, “Cirrhosis can increase portal vein pressure and lead to hypersplenism”.

The author’s answer: We have revised the sentence. The details as follows: “Cirrhosis can increase portal vein pressure and lead to hypersplenism, and splenectomy is often used to treat patients with portal hypertension and hypersplenism.”

5. The two investigators examined the full text, tables, and figures of relevant literature to extract data from articles that had been included in this study.

The author’s answer: We have revised the sentence. The details as follows: “The two investigators examined the full text, tables and figures of relevant literature to extract data from articles that have been included in this study. ”

6. There are some grammatical mistakes in the article please revise the article thoroughly and remove all the errors.

The author’s answer: We have checked the article and fixed some grammatical mistakes.

7. Conclusions are not clearly specified.

The author’s answer: We have revised the conclusions. The details as follows: “Our meta-analysis suggested that combined anticoagulation with LMWH and warfarin can decrease the incidence of PVST in post-splenectomy cirrhotic patients without an increased risk of bleeding. Patients after splenectomy can have short term injections of LMWH followed by long term use of warfarin. And patients were best served with warfarin for 6 months after splenectomy to prevent portal vein thrombosis.”

8. Some of the references need revision such as reference 7 and 19.

The author’s answer: We have revised the references according to the requirements of plos one.

Thank you for your attention and time.

Yours sincerely,

Zheng Liao

Corresponding author:

Name: Jinlong Liu

E-mail: liujl800813@163.com

---

## [Decision Letter · Decision Letter 1]

3 Aug 2023

Long term prophylactic anticoagulation for portal vein thrombosis after splenectomy: A systematic review and meta-analysis

PONE-D-23-13885R1

Dear Dr. Liu,

We’re pleased to inform you that your manuscript has been judged scientifically suitable for publication and will be formally accepted for publication once it meets all outstanding technical requirements.

Kind regards,

Redoy Ranjan, MBBS, MRCSEd, Ch.M., MS (CV&TS), FACS

Academic Editor

PLOS ONE

Additional Editor Comments (optional):

Review Comments to the Author

Reviewer #2: The authors have addressed all the concerns raised. So I recommend accepting the paper in its present form.

Reviewer #3: I think some comments the reviewer suggested were adequately addressed. The manuscript should be accepted.

---

## [Editor Report · Acceptance letter]

7 Aug 2023

PONE-D-23-13885R1 

Long term prophylactic anticoagulation for portal vein thrombosis after splenectomy: A systematic review and meta-analysis 

Dear Dr. Liu:

I'm pleased to inform you that your manuscript has been deemed suitable for publication in PLOS ONE. Congratulations! Your manuscript is now with our production department. 

Kind regards, 

on behalf of

Dr. Redoy Ranjan 

Academic Editor

PLOS ONE